# Risk factors analysis of 90-day mortality in patients with sepsis in intensive care unit

**Yuntian Xu[1], He Zhang[2], Jiezhi Li[3], Nan Wang[4], Huifeng Yuan ⬤[1]\***

1 Department of Critical Care Medicine, The Third Affiliated Hospital of Anhui Medical University, Hefei First People's Hospital, Hefei, China, 2 Department of Emergency Medicine, The First Affiliated Hospital of Anhui Medical University, Hefei, China, 3 Department of Radiology, The First Affiliated Hospital of Anhui Medical University, Hefei, China, 4 Department of Critical Care Medicine, The First Affiliated Hospital of Anhui Medical University, Hefei, China

\* hfyy2020@126.com

## Abstract

### Background

The incidence and mortality of sepsis in the intensive care unit (ICU) remain persistently high. This study primarily investigates the risk factors associated with the 90-day mortality in sepsis patients.

### Method

This retrospective study included 123 sepsis patients admitted to a hospital in China from January 2015 to December 2018, clinical and abdominal CT data were compared between survivors and non-survivors, logistic regression and Cox regression analyses were performed on the abdominal CT data, Finally, survival curves for different skeletal muscle indices were analyzed using the Kaplan-Meier (K-M) method.

### Result

In the abdominal CT scan data, significant differences were observed between survivors and non-survivors in skeletal muscle density (SMD), skeletal muscle area (SMA), skeletal muscle index (SMI), and subcutaneous adipose tissue area (SAT); Cox regression analysis revealed that higher skeletal muscle density (SMD) (HR = 0.953; 95% CI = 0.923–0.984; p = 0.003), skeletal muscle area (SMA) (HR = 0.986; 95% CI = 0.976–0.997; p = 0.011), and skeletal muscle index (SMI) (HR = 0.951; 95% CI = 0.917–0.985; p = 0.005) were significantly associated with lower 90-day mortality compared to non-survivors. Finally, the Kaplan-Meier (K-M) curves demonstrated differences in survival based on the median skeletal muscle index (SMI).

**Data availability statement:** All relevant data are available on Figshare at the following DOI: https://doi.org/10.6084/m9.figshare.28873667. No additional supplementary files are available.

**Funding:** The author(s) received no specific funding for this work.

**Competing interests:** The authors have declared that no competing interests exist.

## Conclusion

Body composition parameters assessed by abdominal CT scans are highly associated with 90-day mortality in ICU patients with sepsis. Among them, SMD, SMA, and SMI are valuable prognostic factors.

## Introduction

Sepsis remains a major cause of mortality among patients in the intensive care unit [1]. Sepsis patients frequently experience muscle catabolism, muscle weakness, and other metabolic dysfunctions, which are classified as sarcopenia or cachexia [2]. Given the prevalence and impact of this condition, the Third International Consensus Definitions for Sepsis and Septic Shock (Sepsis-3) were jointly released in 2016 by the Society of Critical Care Medicine (SCCM) and the European Society of Intensive Care Medicine (ESICM) [3].

Muscular atrophy is a serious sequela of critical illnesses. Low muscle density has been associated with poor prognosis [4]. In particular, muscle loss is associated with increased hospitalization rates and higher mortality among ICU patients. Furthermore, patients who have survived following discharge from the ICU may also experience functional changes that affect their quality of life. Therefore, the concept of sarcopenia has been introduced in 2010 by the European Working Group on Sarcopenia in Older People [5], which is defined as a syndrome characterized by a progressive and comprehensive loss of skeletal muscle mass and strength.

In recent years, studies have shown that skeletal muscle acts as an endocrine organ of cytokines and peptides that plays a key role in the inflamatory process [6]. Also, skeletal muscle content is of great significance to the nutritional status and prognostic risk assessment of ICU patients with sepsis. Given that the systemic inflammatory response represents a risk factor for muscle loss, this warrants further study to define the relationship between skeletal muscle content and survival rate in the presence of inflammatory response [7]. To date, there is no research examining the influence of body composition on the prognostic evaluation of ICU patients with sepsis. Therefore, this study aimed to evaluate the association between body composition and 90-day mortality among these patients.

## Methods

### Patients and data collection

The data for this retrospective study were accessed on **20/12/2024** for research purposes. During and after data collection, the authors did not have access to any information that could identify individual participants. All data were anonymized prior to analysis, ensuring the protection of participant privacy.

This study analyzed 123 sepsis patients who underwent abdominal CT scans and were admitted to the Department of Intensive Care Medicine at Anhui Medical University from January 2015 to December 2018. Sepsis was diagnosed according to the Sepsis-3 criteria, defined as a suspected infection with an acute increase in SOFA score ≥2 points. In this retrospective study, infection was identified based on clinical documentation and the initiation of antibiotic therapy. Microbiological confirmation was not required due to incomplete data availability.

Patient inclusion criteria for the study were 18 years of age or older, hospitalization in the ICU for at least 4 days, and at least one abdominal CT examination within 24 hours before or 4 days after admitting to the ICU. Patients were excluded if they were pregnant or diagnosed with chronic neuromuscular diseases. Patients were excluded from the study if the quality of the CT images were low or the muscle integrity was damaged due to factors such as trauma or artifacts. (Fig 1).

Data including patient age, gender, weight, height, BMI, skeletal muscle density and area, visceral fat area and density, subcutaneous fat area and density, intermuscular fat area and density, the worst scores on APACHE II and SOFA within 24 hours of admission, laboratory indicators, 90-day survival rate, ventilation time, ICU length of stay (ICU-LOS), hospital length of stay, continuous renal replacement therapy (CRRT), Charlson comorbidity index (CCI) [8], and the nutrition risk in critically ill (NUTRIC) score [9] were collected. Telephone follow-up was conducted to gather information on the survival outcomes of patients after discharge.

In-hospital mortality data and other clinical variables, including age, sex, body mass index, SOFA and APACHE II scores, laboratory findings, and comorbidity indexes (CCI, NUTRIC), were extracted from the electronic medical record system. Post-discharge survival status within 90 days was obtained through structured telephone follow-up. These measures ensured a comprehensive dataset for evaluating the association between body composition and 90-day mortality.

## CT scan analysis

High accuracy for the analysis of the body composition of a single-slice abdominal CT scan at the level of the third lumbar vertebra (L3) has been previously demonstrated [10]. Based on the evaluation of magnetic resonance imaging (MRI), the skeletal muscle area at this level is closely related to the volume of skeletal muscle in the whole body [3]. In our study, patients' abdominal CT images were analyzed by experienced radiologists using the Mimics version 17 software (fig 2) [11]. The vertebral bone markers at the L3 level were used to ensure the reproducibility and consistency of measurements.

Varying ranges of HU value (Hounsfield) were used to define body compositions, with the HU of muscle ranging −29 to +150, HU of intermuscular fatty tissue and subcutaneous fat tissue ranging −190 to −30, and HU of visceral fatty tissue ranging −150 to −50 [4,12]. SMD was evaluated by the average radiation muscle attenuation of the entire muscle at the L3 level, and the HU scale was a radioactivity scale that described the tissue density in CT scans [13]. By analyzing the bone density scale, a higher HU value indicated a higher skeletal muscle density and a lesser fat infiltration [14]. Given that the body composition content corresponded to the nutritional status and survival prognosis of patients with sepsis in the ICU, this was analyzed according to the primary endpoint of our study, which was the 90-day mortality rate. (Fig 2).

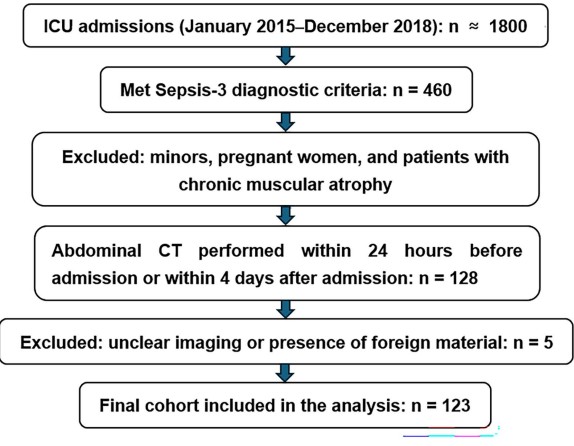

**Fig 1. Patient Selection Flowchart.**

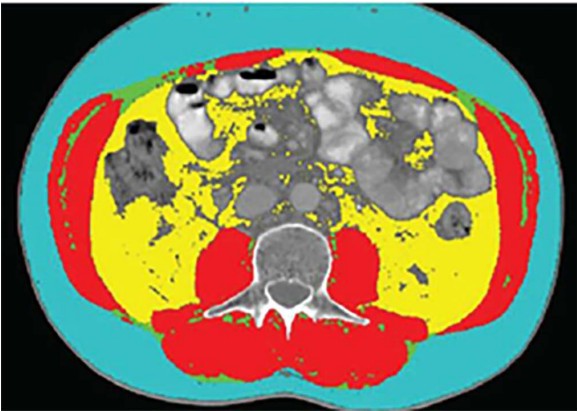

**Fig 2. Example of CT scan showing body composition at the L3 level.** red represents skeletal muscle (SMA), blue represents subcutaneous adipose tissue (SAT), green represents intermuscular adipose tissue (IMAT), and yellow represents visceral adipose tissue (VAT).

## Statistical analysis

A normality test was performed for all the continuous variables. To compare the survival and non-survival groups, the independent sample t-test was used for the normally distributed data, whereas the Mann-Whitney U test was used to analyze non-normally distributed variables. Fisher's exact test and Chi2 test were used to analyze categorical variables. Univariable analyses were first conducted to identify potential predictors of 90-day mortality. Variables with clinical relevance or with $P < 0.10$ in univariate analysis were considered for inclusion in multivariable regression models. Multivariable logistic and Cox proportional hazards regression models were constructed to evaluate the independent association between CT-derived body composition parameters—skeletal muscle density (SMD), skeletal muscle area (SMA), and skeletal muscle index (SMI)—and 90-day mortality. Three models were used:

- **Unadjusted Model**: included only the body composition variable of interest;

- **Model 2**: adjusted for SOFA score, sex, and age;

- **Model 3**: adjusted for SOFA score, sex, age, and Charlson Comorbidity Index (CCI).

All results were reported as odds ratios (OR) or hazard ratios (HR) with 95% confidence intervals (CI). A two-sided $P$-value of < 0.05 was considered statistically significant. The APACHE II score was included in descriptive analyses but was not used in the multivariable models to avoid collinearity with SOFA score and age.

Kaplan-Meier chart was used to analyze the relationship between skeletal muscle index and 90-day mortality (divided into two groups according to the median) and the log-rank test was performed to compare the survival curves between the two groups. The primary outcome of this study was 90-day all-cause mortality. Due to limitations inherent to the retrospective design, data on in-hospital mortality and 30-day mortality were not available and thus were not included in the analysis.

SPSS Statistics 23.0 is used for all the statistical analysis, and values were expressed as mean ± standard deviation (SD) or median and quartile. All data were two-sided. A P-value of <0.05 was considered statistically significant.

## Ethics

The study was approved by the ethics committee of The Third Affiliated Hospital of Anhui Medical University, Hefei First People's Hospital. (No.2024-316-01). Informed consent was waived because the study was retrospective in design. The study was performed in accordance to the Helsinki Declaration. All patients'records were anonymized and de-identified prior to analysis.

## Results

Among the 123 sepsis patients, the 90-day mortality rate was 35.8%. When compared with the survival group, patients in the non-survival group were significantly older (67 ± 16 vs 57 ± 16 years, p = 0.001), had higher APACHE II scores (21 ± 5 vs 17 ± 5, p < 0.001), and higher SOFA scores [10 (8–13) vs 8 (7–11) cm2, p = 0.005] (Table 1). Upon admission to the ICU, the average SMD of all patients was 30.21 HU, the average SMI of all patients was 44.86 cm2/m2, the average SMA was 124.61 cm², and the average SAT was 112.85 cm². The non-survival group had a lower SMD (26.7 ± 9.3 vs 32.2 ± 9.3 HU, p = 0.003), lower SMA (114.8 ± 27.4 vs 130.1 ± 30.9 cm²; p = 0.007), lower SMI (41.6 ± 9.2 vs 46.7 ± 8.9 cm²; p = 0.004), and also lower SAT (96.1 ± 65.4 vs. 122.2 ± 71.6 cm², p = 0.043) when compared with the survival group.

In the unadjusted models, lower values of SMD, SMA, and SMI were significantly associated with increased 90-day mortality. After adjustment for SOFA score, sex, and age in Model 2, all three body composition parameters remained independently associated with reduced mortality (all $P < 0.05$). In Model 3, which additionally included CCI to account for comorbidity burden, the associations persisted with statistical significance, indicating that SMD, SMA, and SMI were robust independent predictors of 90-day mortality in ICU patients with sepsis (Table 2).

COX regression analysis (Table 3) demonstrated that higher SMD, SMA, and SMI was associated with a lower 90-day mortality. After adjusting for the confounding factors, this significant association had persisted (Table 3). –

The Kaplan-Meier curve showed that patients with an SMI lower than the median had a higher mortality rate (Fig 3).

**Table 1. Key patient characteristics by 90-day survival.**

| Variable | Survivors (n = 79) | Non-survivors (n = 44) | P-value |
|---|---|---|---|
| Age (years) | 56.61 ± 16.30 | 67.07 ± 16.23 | 0.001 |
| Male, n (%) | 48 (61%) | 34 (77%) | 0.074 |
| SOFA score (median, IQR) | 8 (7-11) | 10 (8-13) | 0.005 |
| APACHE II score | 16.59 ± 4.86 | 21.36 ± 5.08 | <0.001 |
| Charlson comorbidity index | 3 ± 2 | 6 ± 2 | <0.001 |
| PCT(ng/ml) | 7.54(3.36–38.84) | 7.80(3.89–30.73) | 0.858 |
| SMD (HU) | 32.15 ± 9.26 | 26.74 ± 9.25 | 0.002 |
| SMA (cm²) | 130.09 ± 30.89 | 114.76 ± 27.39 | 0.007 |
| SMI (cm²/m²) | 46.65 ± 8.9 | 41.63 ± 9.17 | 0.004 |
| SAT (cm²) | 122.21 ± 71.56 | 96.06 ± 65.42 | 0.043 |
| Length of ventilation | 6(0-10) | 10(4-16) | 0.002 |
| Hospital length of stay | 22(15-31) | 15(11-23) | 0.005 |
| NA(mg) | 12(0-54) | 80(12-410) | <0.001 |
| CRRT, n (%) | 11(14%) | 15(34%) | 0.012 |

**Table 2. Logistic regression analysis of CT-derived variables and 90-day mortality.**

| Variable | Univariate OR (95% CI), P | Model 2 OR (95% CI), P | Model 3 OR (95% CI), P |
|---|---|---|---|
| SMD (HU) | 0.937 (0.897–0.979), 0.004 | 0.932 (0.881–0.985), 0.013 | 0.950 (0.913–0.989), 0.011 |
| SMA (cm²) | 0.982 (0.969–0.995), 0.009 | 0.965 (0.945–0.985), 0.001 | 0.982 (0.969–0.996), 0.013 |
| SMI (cm²/m²) | 0.937 (0.895–0.981), 0.005 | 0.917 (0.865–0.972), 0.004 | 0.956 (0.916–0.998), 0.040 |
| SAT (cm²) | 0.994 (0.989–1.000), 0.050 | 0.998 (0.991–1.004), 0.494 | 0.998 (0.993–1.003), 0.415 |

Model 2 adjusts SOFA, Age and Gender.

Model 3 adjusts SOFA, Age, Gender and CCI.

**Table 3. Cox regression analysis of CT-derived variables and 90-day mortality.**

| Variable | Univariate HR (95% CI), P | Model 2 HR (95% CI), P | Model 3 HR (95% CI), P |
|---|---|---|---|
| SMD (HU) | 0.953 (0.923–0.984), 0.003 | 0.957 (0.922–0.993), 0.021 | 0.950 (0.913–0.989), 0.011 |
| SMA (cm²) | 0.986 (0.976–0.997), 0.011 | 0.977 (0.964–0.990), 0.001 | 0.982 (0.969–0.996), 0.013 |
| SMI (cm²/m²) | 0.951 (0.917–0.985), 0.005 | 0.943 (0.905–0.982), 0.005 | 0.956 (0.916–0.998), 0.040 |

Model 2 adjusts SOFA, Age and Gender.

Model 3 adjusts SOFA, Age, Gender and CCI.

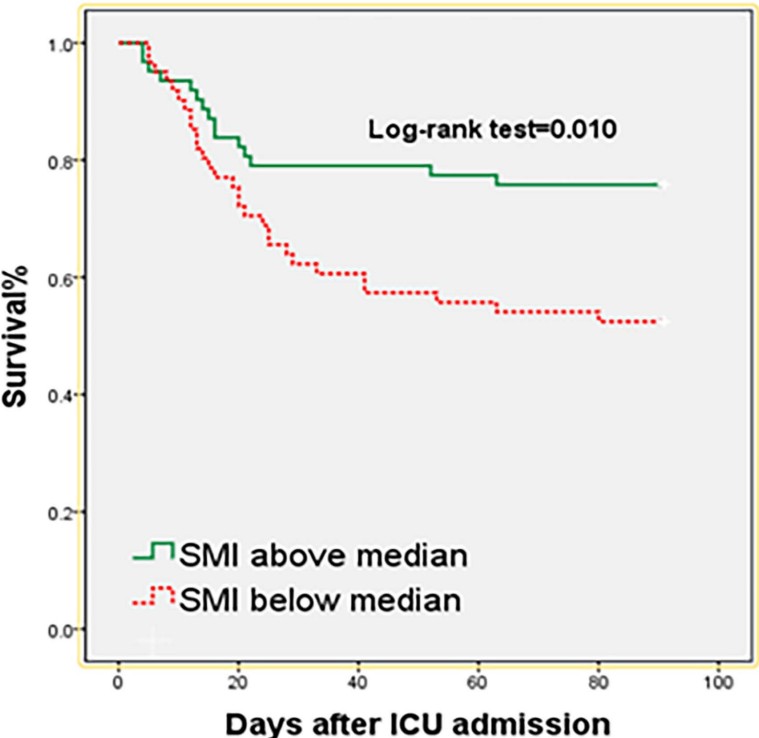

**Fig 3. Kaplan-Meier plots.**

## Discussion

Despite significant advancements in the management of sepsis in the ICU, including early fluid resuscitation and effective antimicrobial therapy, sepsis remains a leading cause of mortality in critically ill patients. Early and dynamic assessment of the severity of sepsis is crucial for improving patient outcomes. The primary indicator currently used in clinical practice to assess the severity and prognosis of sepsis patients is the APACHE II score [15]. APACHE II is a widely used international scoring system for assessing the severity of illness in critically ill patients. It is typically used as a baseline assessment upon ICU admission. The higher the APACHE II score, the more severe the patient's condition, indicating a worse prognosis and higher mortality rate [16]. However, APACHE II lacks an assessment of organ function. The introduction of the SOFA score has addressed this limitation and provides valuable guidance for the management of sepsis patients [17].

Sepsis-related metabolic changes include hypermetabolism, muscle wasting, and fat atrophy, all of which are associated with the inflammatory response [18]. Therefore, markers such as white blood cell count, C-reactive protein, tumor necrosis factor, IL-1, and IL-6 are commonly used as auxiliary diagnostic indicators for sepsis in clinical practice. However, their limitations lie in insufficient sensitivity and specificity [19]. Currently, procalcitonin (PCT) is commonly used in clinical practice as a tool for risk stratification in sepsis. However, non-infectious inflammatory responses such as severe trauma, major surgery, and burns can lead to the release of large amounts of DAMPs (Damage-Associated Molecular Patterns), which in turn can cause an elevation in procalcitonin levels, potentially affecting clinical judgment [20].

Our study has demonstrated that the lower body compositions including the SMA, SMD, and SMI of ICU patients with sepsis are significantly associated with a poorer 90-day overall survival, which is consistent with previous studies and indicating that the severity of skeletal muscle loss is an important predictor of patient prognosis [21–23]. Currently, there are various methods for assessing muscle mass, including bioelectrical impedance analysis, dual-energy X-ray absorptiometry (DXA), magnetic resonance imaging (MRI), and computed tomography (CT) scanning. Compared to bioelectrical impedance analysis and dual-energy X-ray absorptiometry (DXA), CT analysis has the advantage of detecting changes in muscle mass and composition [24]. Additionally, abdominal CT scans, as a routine diagnostic tool, are commonly performed in clinical practice and do not impose any additional burden on patients.

Most sepsis patients often suffer from ICU-acquired weakness, which frequently leads to prolonged mechanical ventilation and extended hospital stays. In the long term, ICU-acquired weakness limits daily activities after discharge and results in a decline in quality of life [25]. The Charlson Comorbidity Index (CCI) is currently the most commonly used tool for assessing comorbidities [8]. Research has shown that higher CCI scores are associated with lower muscle mass, providing important evidence for clinicians in diagnosing sarcopenia [9]. On the other hand, studies have found that higher Nutrition Risk in Critically Ill (NUTRIC) scores are associated with higher 6-month mortality rates [9]. Future rehabilitation and nutritional strategies focused on preventing muscle loss may contribute to better outcomes.

We selected the SOFA score as the primary adjustment variable in multivariable models to account for illness severity, given its clinical relevance in sepsis and lower risk of collinearity. Age and sex were also included due to their established influence on outcomes. In a further model, the CCI was added to control for comorbidities. The associations between skeletal muscle parameters and 90-day mortality remained significant, supporting their independent prognostic value. Although the APACHE II score was significantly higher in non-survivors, it was excluded from the multivariable models due to conceptual and statistical overlap with SOFA score and age. Its inclusion in descriptive comparisons, however, supports its value in characterizing overall disease severity.

Compared to previous studies that primarily focus on short-term outcomes such as in-hospital or 28-day mortality, our study provides valuable insights into the 90-day mortality in sepsis patients, emphasizing the importance of long-term prognosis. Additionally, our research is based on a Chinese ICU cohort, which remains underrepresented in existing literature. These factors contribute to the novelty and clinical relevance of our findings.

## Limitations

This study has several limitations. First, it was a retrospective study conducted at a single tertiary hospital in China, and all included patients were of the same racial background, limiting the generalizability of the findings. Selection bias may also be present, as only patients with sepsis who underwent abdominal CT scans were included. Second, data on nutritional support were not collected, and due to the retrospective design, we were unable to assess patients' nutritional status using the Global Leadership Initiative on Malnutrition (GLIM) criteria. Functional status indicators, such as performance scores or physical function assessments, were also not evaluated. These factors are known to influence prognosis in critically ill patients and may affect mortality outcomes. Third, we did not analyze in-hospital or 30-day mortality due to incomplete follow-up data and variability in hospital length of stay, which are common limitations in retrospective datasets. Although 30-day mortality is often used to assess short-term outcomes, 90-day mortality is also a well-accepted endpoint

that captures longer-term clinical trajectories in patients with sepsis. Future prospective studies incorporating nutritional and functional assessments, as well as short-term mortality outcomes, are warranted to provide a more comprehensive evaluation of sepsis-related risk factors.

## Conclusions

The body composition assessed by the abdominal CT scan is significantly associated with the 90-day survival prognosis of ICU patients with sepsis. Therefore, body composition should be measured and incorporated into the overall assessment of the nutritional status and prognosis in clinical practice.

## Author contributions

**Data curation:** Yuntian Xu, He Zhang, Jiezhi Li, Nan Wang.

**Formal analysis:** Nan Wang.

**Methodology:** Yuntian Xu, Nan Wang, Huifeng Yuan.

**Project administration:** Huifeng Yuan.

**Software:** Jiezhi Li.

**Supervision:** Nan Wang.

**Writing – original draft:** Yuntian Xu.

**Writing – review & editing:** Huifeng Yuan.

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
