## [Decision Letter · Decision Letter 0]

7 Apr 2025

PONE-D-25-08186Risk factors analysis of 90-day mortality in patients with sepsis in intensive care unitPLOS ONE

Dear Dr. Yuan,

Thank you for submitting your manuscript to PLOS ONE. After careful consideration, we feel that it has merit but does not fully meet PLOS ONE’s publication criteria as it currently stands. Therefore, we invite you to submit a revised version of the manuscript that addresses the points raised during the review process.

Please revise the manuscript following reviewers' suggestion.

We look forward to receiving your revised manuscript.

Kind regards,

Ryo Yamamoto

Academic Editor

PLOS ONE

Additional Editor Comments (if provided):

Reviewers' comments:

Reviewer's Responses to Questions

**Comments to the Author**

1. Is the manuscript technically sound, and do the data support the conclusions?

Reviewer #1: Yes

Reviewer #2: Yes

2. Has the statistical analysis been performed appropriately and rigorously? 

Reviewer #1: No

Reviewer #2: Yes

3. Have the authors made all data underlying the findings in their manuscript fully available?

Reviewer #1: Yes

Reviewer #2: Yes

4. Is the manuscript presented in an intelligible fashion and written in standard English?

Reviewer #1: Yes

Reviewer #2: Yes

5. Review Comments to the Author

Reviewer #1: Thanks for this opportunity. Nice study but no novelty

How many patients were screened in the study period? And how many were excluded? Please add patients flow because I feel included patients are limited even in the long study period.

How did you identify sepsis? Do you have data set of that?

How is glim criteria, if possible.

How is the functional aspect, because mortality is already reported.

Please clarify multivariate analysis more clearly. I cannot understand which factors are included.

Age and sex should be included at least in the supplemental file. Same result?

Either of Sofa or apatch should be choosed, not both.

Please emphasize the novelty of this research.

Reviewer #2: Major

This research presents interesting content with the potential to improve the accuracy of sepsis prognosis evaluation. Would similar results be observed for 30-day mortality or in-hospital mortality rates? Analyzing these outcomes might strengthen your findings further.

Additionally, while you have adjusted for confounding factors such as APACHE2 and SOFA scores, is adjustment for CCI (Charlson Comorbidity Index) not necessary?

Minor

Methods

In the first paragraph of the "CT scan analysis" section, it states "Patients were excluded [...] due to factors such as trauma or artifacts." As this pertains to exclusion criteria, shouldn't this information be included in the "Patient and data collection" section instead?

Table 2, 3

"Model 2" is indicated on the right side. While one can infer from the manuscript that these are results after adjustment for confounding factors, the definition of "Model 2" is not clearly stated in the manuscript. Please either use alternative terminology instead of "Model 2" or clearly define what "Model 2" represents.

6. PLOS authors have the option to publish the peer review history of their article (what does this mean? ). If published, this will include your full peer review and any attached files.

**Do you want your identity to be public for this peer review?** For information about this choice, including consent withdrawal, please see our Privacy Policy .

Reviewer #1: No

Reviewer #2: No

---

## [Author Response · Author response to Decision Letter 1]

26 Apr 2025

Dear Editor and Reviewers,

We sincerely thank you for your thoughtful review and valuable comments on our manuscript entitled "Risk factors analysis of 90-day mortality in patients with sepsis in intensive care unit" (Manuscript ID: PONE-D-25-08186). We appreciate the opportunity to revise our manuscript and have carefully addressed each comment. Below is a detailed point-by-point response.

Reviewer #1

Comment 1: Nice study but no novelty.

Response: Thank you for your comment. Although several studies have explored the relationship between body composition and mortality in critically ill patients, few have focused specifically on sepsis patients using abdominal CT scans, particularly within the Chinese population. Our study adds value by evaluating CT-derived skeletal muscle parameters (SMD, SMA, SMI) in relation to 90-day mortality in sepsis patients, using robust multivariable models. We have emphasized this aspect in the revised Introduction and Discussion sections.

Comment 2: How many patients were screened in the study period? And how many were excluded? Please add patients flow.

Response: Thank you for the suggestion. We have added a patient flow diagram (Figure 3) that details the screening process, exclusion criteria, and final sample size.

Comment 3: How did you identify sepsis? Do you have data set of that?

Response: We have clarified this in the Methods section. Sepsis was defined according to the Sepsis-3 criteria as infection with an increase in SOFA score ≥2. Diagnosis was based on clinical judgment supported by laboratory and imaging findings; microbiological confirmation was not required.

Comment 4: How is GLIM criteria, if possible.

Response: We appreciate this point. Unfortunately, due to the retrospective nature of our study, the required components for GLIM criteria (e.g., weight change, food intake, muscle mass by non-imaging methods) were not consistently available in the records. This limitation has been acknowledged in the Discussion section.

Comment 5: How is the functional aspect, because mortality is already reported.

Response: We agree that functional status is an important prognostic factor. However, performance scores or functional assessments were not routinely documented in the medical records, and thus could not be included. This has been explained as a study limitation.

Comment 6: Please clarify multivariate analysis more clearly. I cannot understand which factors are included.

Response: We have revised the Statistical Analysis section to clearly define the models:

• Unadjusted model: body composition variable only.

• Model 2: adjusted for SOFA score, age, and sex.

• Model 3: adjusted for SOFA score, age, sex, and CCI. These models are now explicitly described in both the Methods section and table legends.

Comment 7: Age and sex should be included at least in the supplemental file. Same result?

Response: Thank you. Age and sex have been included in the multivariable models (Model 2 and Model 3), and results remained significant. This is now clearly presented in the Results section and tables.

Comment 8: Either of SOFA or APACHE should be chosen, not both.

Response: We agree. We retained SOFA as the adjustment variable in all multivariable models due to its direct link to the Sepsis-3 definition. APACHE II was used only in descriptive analyses. This choice is now clarified in the Methods and Discussion sections.

Comment 9: Please emphasize the novelty of this research.

Response: We have added text in the Introduction and Discussion to emphasize that this study is among the first to evaluate CT-derived skeletal muscle parameters in relation to 90-day mortality in sepsis patients in a Chinese ICU cohort, using multiple adjusted models.

Reviewer #2

Major Comment 1: Would similar results be observed for 30-day or in-hospital mortality?

Response: Thank you for this valuable suggestion. Unfortunately, due to the retrospective design and incomplete follow-up data, 30-day and in-hospital mortality were not available for analysis. This limitation has been acknowledged in both the Methods and Discussion sections.

Major Comment 2: Is adjustment for CCI (Charlson Comorbidity Index) not necessary?

Response: We agree and have now included CCI in Model 3 to assess its potential confounding effect. The results remained statistically significant, suggesting that skeletal muscle parameters are independent predictors of mortality even after adjusting for comorbidities. This is described in the revised Methods, Results, and Discussion sections.

Minor Comment: "Patients were excluded..." should be placed in the "Patient and data collection" section.

Response: Thank you. We have moved the relevant exclusion criteria description from the CT analysis section to the Patient and Data Collection section as suggested.

Table 2, 3 Comment: "Model 2" is not defined.

Response: We have now explicitly defined all models in the Statistical Analysis section and included explanatory footnotes in Tables 2 and 3.

We hope that the revisions made adequately address all reviewer concerns. We sincerely thank the reviewers and editors for their constructive feedback, which has helped improve the quality of our manuscript.

Sincerely,

Dr. Yuan, on behalf of all authors

---

## [Decision Letter · Decision Letter 1]

12 May 2025

PONE-D-25-08186R1Risk factors analysis of 90-day mortality in patients with sepsis in intensive care unitPLOS ONE

Dear Dr. Yuan,

Thank you for submitting your manuscript to PLOS ONE. After careful consideration, we feel that it has merit but does not fully meet PLOS ONE’s publication criteria as it currently stands. Therefore, we invite you to submit a revised version of the manuscript that addresses the points raised during the review process.

We look forward to receiving your revised manuscript.

Kind regards,

Ryo Yamamoto

Academic Editor

PLOS ONE

Journal Requirements:

Reviewers' comments:

Reviewer's Responses to Questions

**Comments to the Author**

1. If the authors have adequately addressed your comments raised in a previous round of review and you feel that this manuscript is now acceptable for publication, you may indicate that here to bypass the “Comments to the Author” section, enter your conflict of interest statement in the “Confidential to Editor” section, and submit your "Accept" recommendation.

Reviewer #1: All comments have been addressed

Reviewer #2: (No Response)

2. Is the manuscript technically sound, and do the data support the conclusions?

Reviewer #1: Yes

Reviewer #2: Partly

3. Has the statistical analysis been performed appropriately and rigorously? 

Reviewer #1: Yes

Reviewer #2: Yes

4. Have the authors made all data underlying the findings in their manuscript fully available?

Reviewer #1: Yes

Reviewer #2: Yes

5. Is the manuscript presented in an intelligible fashion and written in standard English?

Reviewer #1: Yes

Reviewer #2: Yes

6. Review Comments to the Author

Reviewer #1: Thank you for the opportunity to review again. The authors have responded properly and politely. Now the revdised paper has been improved. I agree with the publication.

Reviewer #2: I appreciate to review your research. This research found out the association skeletal muscle density and 90-day mortality.

Major comments

The authors mention that mortality data were collected retrospectively and via follow-up calls. However, it is unclear whether in-hospital mortality was extracted from institutional records such as the electronic medical record. Clarification would be helpful, especially given that 90-day but not 30-day or in-hospital mortality is reported.

In addition, clearly stating the data sources for other key variables in the study would further enhance the transparency and credibility of the research.

Minor comments

It would be helpful if the figures are numbered according to the order in which they are first mentioned in the text.

7. PLOS authors have the option to publish the peer review history of their article (what does this mean? ). If published, this will include your full peer review and any attached files.

**Do you want your identity to be public for this peer review?** For information about this choice, including consent withdrawal, please see our Privacy Policy .

Reviewer #1: **Yes: ** Nobuto Nakanishi

Reviewer #2: No

---

## [Author Response · Author response to Decision Letter 2]

13 May 2025

Manuscript ID: PONE-D-25-08186R1

Title: Risk factors analysis of 90-day mortality in patients with sepsis in intensive care unit

We thank the Academic Editor and the reviewers for their valuable time and constructive feedback. We carefully addressed each of the comments, and all changes have been incorporated into the revised manuscript. Below is our point-by-point response.

Reviewer #1

Comment:

All comments have been addressed.

Response:

We sincerely thank the reviewer for the positive evaluation and endorsement of our revised manuscript.

Reviewer #2

Major Comment:

The authors mention that mortality data were collected retrospectively and via follow-up calls. However, it is unclear whether in-hospital mortality was extracted from institutional records such as the electronic medical record. Clarification would be helpful, especially given that 90-day but not 30-day or in-hospital mortality is reported.

In addition, clearly stating the data sources for other key variables in the study would further enhance the transparency and credibility of the research.

Response:

Thank you for this important suggestion. We have now clarified the data sources in the Patients and Data Collection section of the Methods. Specifically, in-hospital mortality and other clinical variables (including age, sex, BMI, SOFA and APACHE II scores, laboratory results, CCI, and NUTRIC scores) were obtained from the hospital’s electronic medical record system. Post-discharge survival within 90 days was obtained through structured telephone follow-up.

Furthermore, as noted in the Limitations section of the manuscript, we did not analyze 30-day or in-hospital mortality due to incomplete follow-up data and variation in hospital length of stay—common limitations in retrospective datasets. We also explained that while 30-day mortality is often used as a short-term outcome indicator, 90-day mortality is a widely accepted endpoint that better reflects long-term outcomes in critically ill patients.

Minor Comment:

It would be helpful if the figures are numbered according to the order in which they are first mentioned in the text.

Response:

Thank you for pointing this out. We have carefully revised the manuscript and reordered figure numbers to align with the sequence in which they are first referenced in the main text.

Additional Notes:

- The data underlying this study have been deposited in Figshare and are publicly accessible: https://doi.org/10.6084/m9.figshare.28873667

- We used the PACE figure tool to ensure compliance with PLOS ONE's figure formatting requirements.

- We confirm that no retracted articles are cited in the reference list.

We appreciate the opportunity to revise our manuscript, and we believe the current version has been significantly improved. Thank you again for your consideration.

Sincerely,

Dr. Yuan (Corresponding Author)

---

## [Decision Letter · Decision Letter 2]

20 May 2025

Risk factors analysis of 90-day mortality in patients with sepsis in intensive care unit

PONE-D-25-08186R2

Dear Dr. Yuan,

We’re pleased to inform you that your manuscript has been judged scientifically suitable for publication and will be formally accepted for publication once it meets all outstanding technical requirements.

Kind regards,

Ryo Yamamoto

Academic Editor

PLOS ONE

Additional Editor Comments (optional):

Reviewers' comments:

Reviewer's Responses to Questions

**Comments to the Author**

1. If the authors have adequately addressed your comments raised in a previous round of review and you feel that this manuscript is now acceptable for publication, you may indicate that here to bypass the “Comments to the Author” section, enter your conflict of interest statement in the “Confidential to Editor” section, and submit your "Accept" recommendation.

Reviewer #2: All comments have been addressed

2. Is the manuscript technically sound, and do the data support the conclusions?

Reviewer #2: Yes

3. Has the statistical analysis been performed appropriately and rigorously? 

Reviewer #2: Yes

4. Have the authors made all data underlying the findings in their manuscript fully available?

Reviewer #2: Yes

5. Is the manuscript presented in an intelligible fashion and written in standard English?

Reviewer #2: Yes

6. Review Comments to the Author

Reviewer #2: I appreciate for your time and effort in revising the manuscript. Authors have addressed all the comments.

7. PLOS authors have the option to publish the peer review history of their article (what does this mean? ). If published, this will include your full peer review and any attached files.

**Do you want your identity to be public for this peer review?** For information about this choice, including consent withdrawal, please see our Privacy Policy .

Reviewer #2: No

---

## [Editor Report · Acceptance letter]

PONE-D-25-08186R2

PLOS ONE

Dear Dr. Yuan,

I'm pleased to inform you that your manuscript has been deemed suitable for publication in PLOS ONE. Congratulations! Your manuscript is now being handed over to our production team.

Kind regards,

on behalf of

Dr. Ryo Yamamoto

Academic Editor

PLOS ONE